# A fast MR-thermometry method for quantitative assessment of temperature increase near an implanted wire

Marylène Delcey[1,2,3,4]*, Pierre Bour[1,2,3], Valéry Ozenne[1,2,3], Wadie Ben Hassen[4], Bruno Quesson[1,2,3]

**1** IHU Liryc, Electrophysiology and Heart Modeling Institute, Fondation Bordeaux Université, Pessac-Bordeaux, France, **2** Centre de recherche Cardio-Thoracique de Bordeaux, U1045, Univ. Bordeaux, Bordeaux, France, **3** INSERM, Centre de recherche Cardio-Thoracique de Bordeaux, U1045, Bordeaux, France, **4** Siemens Healthcare, Saint-Denis, France

* marylene.delcey@ihu-liryc.fr

## Abstract

**Data Availability Statement:** Data has been uploaded to Zenodo: https://zenodo.org/record/4735614 (DOI: 10.5281/zenodo.4735614).

### Purpose

To propose a MR-thermometry method and associated data processing technique to predict the maximal RF-induced temperature increase near an implanted wire for any other MRI sequence.

### Methods

A dynamic single shot echo planar imaging sequence was implemented that interleaves acquisition of several slices every second and an energy deposition module with adjustable parameters. Temperature images were processed in real time and compared to invasive fiber-optic measurements to assess accuracy of the method. The standard deviation of temperature was measured in gel and in vivo in the human brain of a volunteer. Temperature increases were measured for different RF exposure levels in a phantom containing an inserted wire and then a MR-conditional pacemaker lead. These calibration data set were fitted to a semi-empirical model allowing estimation of temperature increase of other acquisition sequences.

### Results

The precision of the measurement obtained after filtering with a 1.6x1.6 mm$^2$ in plane resolution was 0.2˚C in gel, as well as in the human brain. A high correspondence was observed with invasive temperature measurements during RF-induced heating (0.5˚C RMSE for a 11.5˚C temperature increase). Temperature rises of 32.4˚C and 6.5˚C were reached at the tip of a wire and of a pacemaker lead, respectively. After successful fitting of temperature curves of the calibration data set, temperature rise predicted by the model was in good agreement (around 5% difference) with measured temperature by a fiber optic probe, for three other MRI sequences.

**Funding:** This work received the financial support from the French National Investments for the Future Programs ANR-10-IAHU-04 (IHU Liryc), the Laboratory of Excellence ANR-10-LABX-57 (TRAIL, BQ), the French National Investment: ANR-17-CE19-0007(CARTLOVE, VO), and the French National Investment: ANR-19-CE19-0008-01 (CARCOI, BQ). The funders had no role in study design, data collection and analysis, decision to publish, or preparation of the manuscript. The funder provided support in the form of salaries for authors [MD and WBH], but did not have any additional role in the study design, data collection and analysis, decision to publish, or preparation of the manuscript.The specific roles of these authors are articulated in the 'author contributions' section.

## Conclusion

This method proposes a rapid and reliable quantification of the temperature rise near an implanted wire. Calibration data set and resulting fitting coefficients can be used to estimate temperature increase for any MRI sequence as function of its power and duration.

## Introduction

Magnetic resonance imaging (MRI) is increasingly performed in the presence of cardiac electronic implantable devices (CEIDs) [1, 2], or deep brain stimulation (DBS) electrodes [3], together with interventional devices such as for MRI-guided catheterization [4]. In these situations, radiofrequency pulses of the MRI sequence are considered as a principal risk since energy deposition in the patient may induce currents along the device's conductive part and result in local hotspots at its interface with the surrounding tissue [5]. Several theoretical [6–8] and experimental studies [9–12] have shown that tissue temperature increase can easily reach several tens of degrees Celsius, potentially leading to severe burn injuries [13, 14].

Standards were imposed by the U.S Food and Drug Administration (FDA) and the International Electrotechnical Commission (IEC) [15] both for maximum tissue temperature and RF exposure conditions. Various methods have been proposed to evaluate device safety [16]. Numerical simulations of electromagnetic fields, induced currents and resulting temperature evolution (using the bio-heat transfer equation) [17–19] require precise knowledge of the device's 3D geometrical arrangements, its composition, size and its position relative to the MRI scanner's excitation coil, together with electrical and thermal tissue properties, making personalized simulation for each patient unpractical. To overcome this limitation, in vivo assessments are preferred. One solution is to integrate temperature sensors in the device, located where hot spots are likely to occur [11]. However, these methods require transmission of temperature readings during MRI scanning which complicates the design of the device itself, particularly for DBS and CEIDs where no percutaneous access to the device is available. Moreover, in cases of disconnection or rupture of the device lead [20], the location of the sensor may no longer correspond to the anticipated hot spot location in surrounding tissue. Other approaches based on MRI measurements were proposed to quantify induced currents into the device through $B_{1+}$ mapping, using either magnitude [21] or phase images [22]. However, these methods still require some assumptions (homogeneous or known B1 transmit and/or receive fields) to work efficiently. Moreover, they suffer from being simple surrogates of the relevant quantity of interest which is tissue temperature. In the context of measuring small temperature changes near devices, MRI-thermometry should be rapid with a sufficiently large spatial coverage around the implanted wire and provide a spatial resolution in the range of a few millimeters. The thermometry method should also be dynamic to monitor temperature evolution, while being precise enough to map small temperature changes with degree of uncertainty below 1˚C. MRI-Temperature imaging methods based on Proton Resonance Frequency Shift (PRFS) [23, 24], T1-measurements [25–27] and using paramagnetic lanthanide complex [28] have been proposed. However, they do not fulfil all requirements in terms of spatial resolution, rapidity, spatial coverage and precision of temperature measurements specified above. The temperature rise near the tip of an implanted wire depends on the local absorbed power by the tissue and its conversion into heat. The maximal temperature reached at the end of a MRI sequence varies depending on tissue thermal diffusivity and perfusion. Thus, time-

average values such as SAR or $B_{1+rms}$ and total energy emitted by the sequence may be insufficient to predict maximal temperature rise near an implanted wire for any MRI sequence.

In this study, we propose a method to address these points in the context of implanted wires. A sub-second dynamic MRI-thermometry method based on the PRFS technique was implemented, including a module for energy deposition interleaved between successive rapid temperature measurements. We measured the local temperature increase with this sequence near the tip of an implanted wire into a gel for various acquisition conditions, to create a calibration data set. We also propose a model for fitting temperature evolution of this calibration data set together with an associated processing method to predict the maximal temperature rise for other MRI sequences.

## Materials and methods

### Set-up for ex vivo experiments

A Plexiglas box filled with agar (2% with 0.9% NaCl to match tissue electrical conductivity) was used for experiments. The container was designed to position a copper wire vertically and to hold a fluoroptic probe perpendicular to the wire.

### MRI-thermometry sequence

All measurements were performed on a 1.5T clinical imaging system (MAGNETOM Avanto-fit, Siemens Healthcare Erlangen, Germany) equipped with a maximum gradient strength of 45 mT/m and a maximum slew rate of 200 T/m/s. A circular loop of 11 cm in diameter and two spine elements (4 elements each) were used for imaging (for a total of 9 receiver coils). The acquisition sequence (Fig 1) was a modified single-shot gradient echo planar imaging (EPI) sequence, with the following parameters: FOV = 120 mm, Repetition Time(TR)/Echo Time(TE) = 1000/18 ms, matrix size = 74x74 pixels (zero filled to 148x148 pixels), slice thickness = 2.4 mm, bandwidth = 1648 Hz/pixel, FA = 53˚, GRAPPA acceleration factor of 2, 7/8 Partial Fourier. Between each EPI acquisition (62 ms per slice, including fat saturation pulses), a train of RF-pulses (called "heating module" in the remaining text) was applied between dynamic acquisitions #10 and #90 with adjustable parameters: flip angle, inter-pulse delay and number of pulses. In the remaining text, the flip angle of the heating module RF pulses is called $FA_{HM}$. Each RF pulse had a sinc shape of 1 ms duration (with an inter-pulse delay of 2 ms) and was emitted with a tunable frequency offset (typically 100 kHz) to avoid direct proton signal

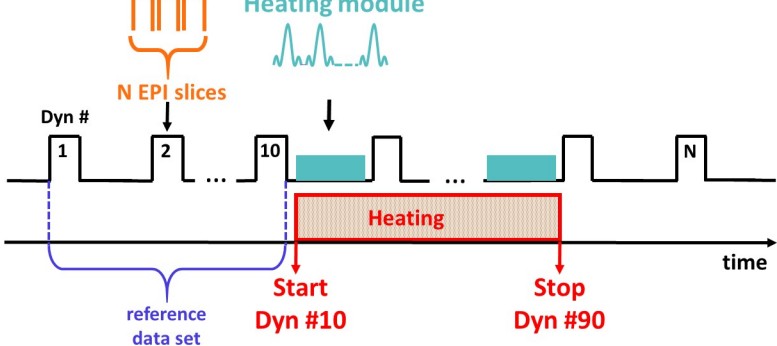

**Fig 1. Schematic of the MRI-thermometry technique.** Single shot gradient echo EPI acquisition interleaved with a train of RF pulses (heating module) with adjustable flip angle, inter-pulse delay and number of pulses.

saturation [24, 26]. For 3 EPI slices and a TR of 1 s, 242 pulses were played between each stack of slices, resulting in a duty cycle of the heating module of 72% per TR.

## Thermometry pipeline

Temperature evolution was computed and visualized in real time during experiments using an MRI-thermometry pipeline similar to one proposed for monitoring cardiac radiofrequency ablations [29, 30]. The MRI raw data were streamed through TCP/IP to the Gadgetron framework for online image reconstruction [31], including EPI ghost-correction followed by GRAPPA reconstruction [32]. Prior to Fourier transform of the data, zero filling was applied resulting in a matrix size of 148x148 pixels and a reconstructed pixel spacing of $0.8x0.8mm^2$. Temperature images were then computed from phase images using the PRFS method (with a constant of $-0.0094$ ppm/˚C) [33, 34]. The first 10 acquired slice stacks in the time series were averaged together to create reference phase images for each slice. Potential spatio-temporal phase-drifts were corrected using the method proposed by Ozenne et al. [29], using a temporal sliding window over the last 10 acquired stacks. Finally, a low pass temporal filter (first order Butterworth with 0.04 Hz cutoff frequency) was applied on a pixel-by-pixel basis on temperature curves to reduce uncertainty. Resulting temperature maps were sent online to a remote computer for display (Thermoguide, Image Guided Therapy, Pessac, France).

## $B_{1+rms}$ measurements

$B_{1+rms}$ values were dynamically retrieved from the MRI scanner interface during acquisition at dynamic #90 (end of heating module). The total energy emitted by the sequence was computed and displayed in the user interface of the MRI console. A minimum delay of 6 minutes was observed between consecutive measurements with a different $FA_{HM}$ to reset $B_{1+rms}$ values by the MRI console. This delay also ensured proper cooling of the gel between consecutive experiments.

## Validation of the MRI-thermometry method

For validation purposes, a test experiment was performed in a gel containing a copper wire (0.4 mm diameter, 1.2m length). One end of the wire was inserted vertically into the Plexiglas tank filled with agar gel. The remaining part of the wire was positioned in contact with the tunnel bore to favor RF-induced heating (highest electric field emitted by the transmit coil [6]). A fluoroptic temperature fiber (Luxtron® Fiber Optic, STF probe, LumaSense Technologies, Santa Clara, CA, USA) was inserted in the gel perpendicularly to the copper wire. The distance between the wire tip and the optical sensor was approximately 1 mm. A 3D balanced-SSFP sequence was acquired to locate the fiber optic temperature probe within the gel, using the following acquisition parameters: bandwidth = 250 Hz/pixel, TR/TE = 666/2.43 ms, 0.8 mm isotropic resolution, FOV = 130 mm, Flip Angle = 90˚. The position of the optical fiber tip was identified and the slice stacks of the thermometry sequence were positioned at this reference location.

## Potential RF-induced heating near the implanted wire measurement

In a second batch of experiments, another gel of identical content was used and the optical fiber temperature sensor was not inserted to obtain temperature maps devoid of any signal drop close to the wire. The same imaging sequence was repeated while varying the $FA_{HM}$ from 0˚ to 90˚ by steps of 10˚ in order to create a calibration dataset. For each acquisition, the temperature evolution in the same pixel was analyzed, selecting the pixel with the maximal

temperature increase at the end of the energy deposition (acquisition #90) from the temperature data corresponding to the largest $FA_{HM}$. To verify absence of temperature drift during experiment, temperature evolution in a pixel located outside the heating zone was also plotted.

## Potential RF-induced heating near a pacemaker lead

We evaluated our method on a commercial MR conditional pacemaker lead (CapSureFix Novus MRI Surescan, 65-cm length, Medtronic). The latter was inserted vertically (perpendicular to $B_0$) into a gel and not connected to its generator to simulate an abandoned lead scenario. The tip of the lead that is normally screwed into the myocardium was inserted into the gel while the other extremity was left in the air. A 3D gradient echo (TE/TR = 3.9/8 ms, isotropic resolution of 0.8 mm) was acquired to locate the lead and position the central slice (stack of 3 slices) of the proposed sequence (with acquisition parameters identical to those mentioned above) at the lead tip.

## Statistical analyses

To assess the thermometry precision, a first acquisition with $FA_{HM} = 0°$ of the heating module was performed in gel. The same scan parameters as described in section MRI-thermometry sequence were used. The temporal average of temperature ($\mu_T$) and the temporal standard deviation of temperature ($\sigma_T$) were computed for each pixel in a region of interest around the wire over the 120 dynamic acquisitions. The same analysis was repeated after temporal filtering.

## MRI-thermometry assessment in volunteer

A healthy volunteer was informed about the protocol and consented to be included in the study (the institution review board "Comité de protection des personnes îles de France IV" #IRB0003835 approved this study under the approval number 2017-A03313-50) in order to measure the mean temporal standard deviation of the temperature in the brain with the proposed method, without energy deposition ($FA_{HM} = 0°$). Image acquisition parameters were 40° FA, 149*149 mm FOV, 92x92 matrix (zero filled to 184x184), 1510 Hz/px bandwidth, 70 repetitions, 1s repetition time. Measurements were repeated with different TE values of 22, 30, 40, 50, 60 and 70 ms. The standard 16-elements head coil provided by the manufacturer was used. A ROI was manually drawn to cover most of the brain over the 3 slices. The temporal standard deviation ($\sigma_T$) was computed over the 3 slices and analyzed with a Box-and-Whisker plot (selected values: lower value, first quartile, median value, third quartile and 95% of the distribution) to characterize precision of the method.

## Temperature dependence on flip angle, $B_{1+rms}$ and energy emitted by the MRI sequence

For each experiment of the calibration dataset, a temporal window of 5 dynamic acquisitions was used to compute the mean temperature and the temporal standard deviation at the end of energy deposition (between acquisitions #86 and #90). The $\mu_T \pm \sigma_T$ temperature values were plotted as a function of the flip angle, $B_{1+rms}$ and energy emitted (i.e. sum of the energies of each individual RF pulse, including pulses for imaging and pulses of the heating module). A quadratic fit was performed on the resulting first two curves and a linear fit on the last one. Coefficients (namely, $\beta_1$, $\beta_2$ and $\beta_3$) and $R^2$ of the fit were retrieved.

### Prediction of temperature increase for other MRI sequences

In this section, we propose a semi-empirical approach to exploit temperature data obtained from a calibration dataset to predict the maximal temperature rise for any other MRI sequence. Considering that heating induced near an implanted wire is localized around its tip, we chose to approximate this heating source by a Gaussian function with isotropic dimensions. Under this assumption, temperature evolution at the hottest point resulting from energy deposition at constant power ($P_0$) applied between $t_0$ and $t_1$ can be analytically described by the following equation [35]:

$$T(t) = \begin{cases} 0 & \text{for } t \leq t_0 \\ \alpha \, P_0 \, \tau \ln \dfrac{t - t_0 + \tau}{\tau} & \text{for } t_0 \leq t \leq t_1 \\ \alpha \, P_0 \, \tau \ln \dfrac{t - t_0 + \tau}{t - t_1 + \tau} & \text{for } t \geq t_1 \end{cases} \tag{1}$$

Where $\alpha$ is the absorption coefficient and $\tau$ is a time constant. Temperature evolution at the hottest point for each temperature curve of the calibration data set acquired at different powers $P_i$ (i.e. for each flip angle of the SAR module) was fit using equation [1] to retrieve $\alpha$ and $\tau$. Then, we plot $\alpha_i$ and $\tau_i$ as a function of $P_i$ and fit these two curves with a second order polynomial function. The resulting functions allow then to compute $\alpha$ and $\tau$ values corresponding to the power of any other MRI sequence. Thus, temperature evolution for the selected sequence can then be simulated by taking its effective emitted power (total energy divided by acquisition duration) and its acquisition duration. In a third batch of experiments, we included the tip of the wire already described above together with the optical fiber into a gel (wire perpendicular to $B_0$ and identical gel preparation as described above). After the calibration data set was created, temperature curves for each flip angle were processed as indicated above. Then, three other acquisition sequences typically used in clinic were launched and temperature was recorded by the fiber optic probe:

- A 2D Turbo spin echo sequence, emitting 11.093 W power during 38 s

- A 3D gradient echo sequence, emitting 2.522 W during 2 min 15 s

- A 2D cine true-fisp sequence, emitting 43.590 W during 9s

Temperature evolution simulated for these 3 sequences using equation [1] and parameters derived from the proposed method were compared to fiber optic readings.

## Results

### Precision of the MRI-thermometry method

Fig 2A shows the temporal average of temperature ($\mu_T$) and the temporal standard deviation of temperature ($\sigma_T$) in the gel over the 120 dynamic acquisitions for the first slice, when no energy is deposited ($FA_{HM} = 0°$ for the heating module). $\mu_T$ and $\sigma_T$ values were (mean ± std) 0.00±0.20°C and 0.65±0.05°C without filtering, and 0.00±0.20°C and 0.21±0.04°C after filtering, respectively. Fig 2B shows the three slices acquired on a volunteer with the proposed method (TE = 30 ms) together with the map of temporal standard deviation in an ROI covering most of the brain. Box-and-whisker plots of $\sigma_T$ show that median values decreased from 0.2°C for a TE of 22 ms to 0.12°C for a TE ranging 40–70 ms. Moreover, at least 75% of the pixels included in the ROI remained below 0.25°C, irrespective of the echo time (90% or more for TE ranging 40–70 ms).

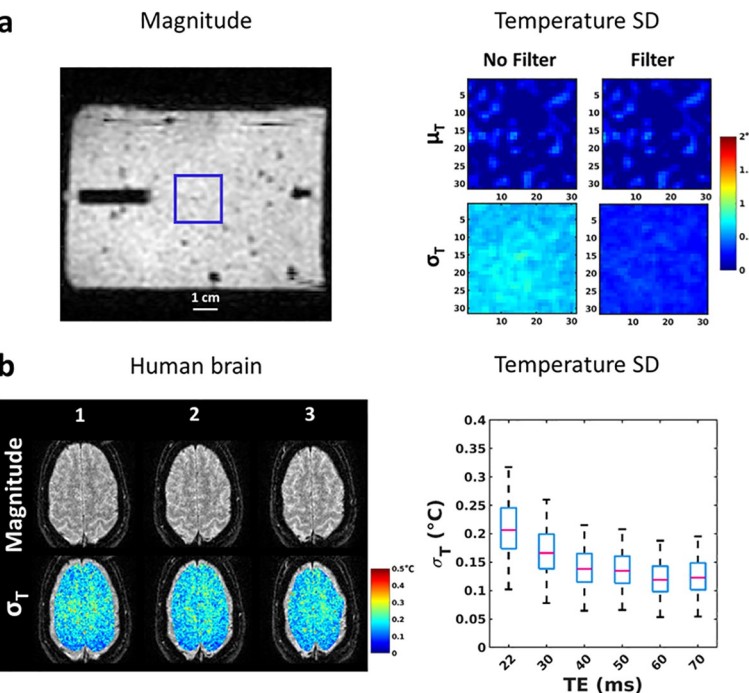

**Fig 2. Temperature precision in gel and in human brain.** (a) Left: Magnitude image of the thermometry sequence. The overlaid blue square delimits the region where the analysis of temperature data was performed. Horizontal bar represents 1 cm. Right: Maps of $\mu_T$ and $\sigma_T$ computed over the complete time series before and after filtering with a Butterworth low-pass filter. Mean ± SD of $\mu_T$ and $\sigma_T$ were 0.00±0.2˚C and 0.65±0.05˚C before filtering and 0.00±0.2˚C and 0.21±0.04˚C after filtering, respectively. (b) Left: Measurement of the temperature standard deviation over the brain of a healthy volunteer. Images on the left show the magnitude images (top row) averaged over 10 consecutive acquisitions and temporal standard deviation of temperature ($\sigma_T$, bottom row) for a TE of 30 ms in a large ROI covering the brain. Right: Box and whiskers plots show the distribution of $\sigma_T$ for different TE within the ROI. Median values are displayed in pink and box correspond to 25% (bottom of the blue box) and 75% (top of the blue box) of the distribution, while the upper limit of the whiskers corresponds to 95% of the pixels in the ROI.

## Accuracy of the MRI-thermometry method during heating

Fig 3A shows the magnitude image of the gel sample with the optical fiber inserted near the wire tip. Fig 3B displays the temperature distribution at the end of the energy deposition

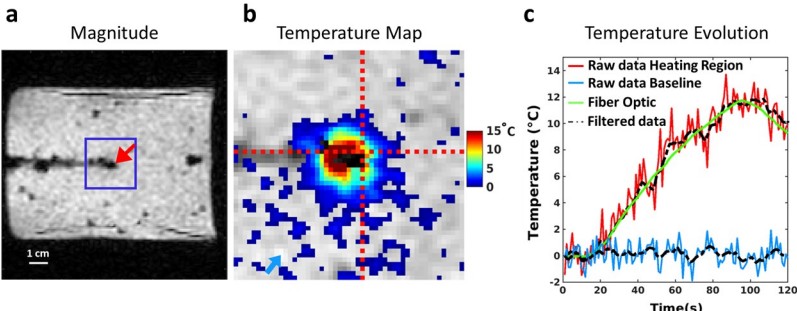

**Fig 3. Comparison between temperature values measured by the optical fiber and the proposed imaging method.** (a) Magnitude image where the fluoroptic sensor is visible. The blue square represents the region of interest and red arrow indicates the location of the fluoroptic tip. Horizontal bar represents 1 cm. (b) Zoomed view of temperature map overlaid on magnitude image at the end of heating (dynamic acquisition #90). Intersection between the dashed red lines shows the pixel corresponding to the optical fiber tip location. The blue arrow indicates the selected pixel located outside the heated region. (c) Temperature evolution (red and blue curves) plotted for the selected pixels in image (b) with the temperature curve obtained from the optical fiber (green). Dashed lines are filtered curves.

(dynamic acquisition #90) within the blue square shown in Fig 3A. Local heating can be observed around the tip of the copper wire. Evolution of the temperature over the 120 dynamic acquisitions is plotted in Fig 3C for a single pixel located near the fiber optic sensor, together with temperature evolution in another pixel located away from the heated region. Overlaid dashed lines correspond to the MRI-temperature data in the same pixels after low-pass filtering. A strong correspondence is observed between temperature evolution measured by the fiber optic sensor (green curve) and filtered MRI-temperature data (dashed black curve). The maximal temperature value computed over 5 dynamic acquisitions around #90 for filtered MRI-thermometry data and fluoroptic probe were 11.5˚C and 11.7˚C, respectively. To compensate the latency induced by the filter (delay of three repetition times) and compute correct root mean squared error (RMSE) values, the filtered curve was shifted left by three dynamic acquisitions in post processing before subtraction to temperature readings from the optical thermometer. The resulting RMSE were 1.2˚C and 0.5˚C for unfiltered and filtered MR temperature values, respectively.

## Phantom experiments with varying flip angles

Fig 4A displays the MRI-temperature maps at the dynamic acquisition #90 for each flip angle of the heating module. No artifact related to the presence of the wire was observed on the magnitude image of the thermometry sequence. A temperature increase was observed close to the tip of the wire, with an increasing maximal value with the flip angle. In the present configuration, the maximal temperature increase was 32.4˚C for a 90˚ flip angle. Temperature evolution is plotted in Fig 4B (red curves) for each flip angle in the same pixel (intersection of the dashed red lines) and in a pixel located outside the heated region (blue curves). Table 1 reports the measured $B_{1+rms}$, total emitted energy and maximal temperature increases for flip angles of the

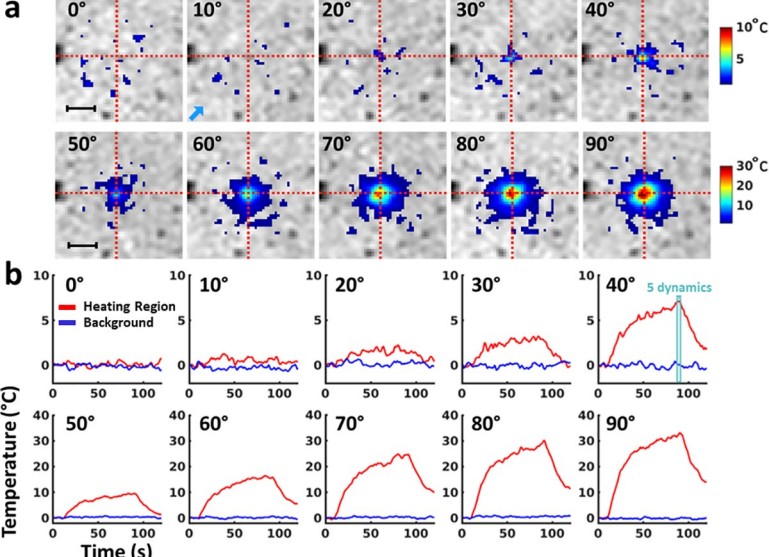

**Fig 4.** Evolution of temperature distribution with increasing flip angle: (a) unfiltered temperature maps are overlaid on their corresponding cropped magnitude images at the end of heating (acquisition #90) and show the temperature spatial distribution for each flip angle. Intersection of red lines indicates the pixel of interest and the blue arrow indicates the pixel selected for background, respectively. This pixel is the same for every acquisition and is located in the region of maximum heating. Horizontal bar in the top left images represents 1 cm. (b) Temperature evolution versus time for the pixel of interest. Red and blue curves show the temperature evolution in the pixel of interest and in a pixel outside the region of interest respectively.

**Table 1. Summary of experimental conditions and temperature increase as a function of the flip angle of the heating module.**

| FA (°) (heating module) | $B_{1+rms}$ (µT) (EPI + heating module) | Energy (J) (heating module) | Max temperature increase (°C) |
|---|---|---|---|
| 0 | 0.6 | 0 | 0.2 |
| 10 | 0.8 | 150 | 0.7 |
| 20 | 1.1 | 602 | 1.4 |
| 30 | 1.4 | 1354 | 2.6 |
| 40 | 1.9 | 2407 | 6.7 |
| 50 | 2.3 | 3761 | 9 |
| 60 | 2.8 | 5415 | 15.7 |
| 70 | 3.2 | 7371 | 24 |
| 80 | 3.8 | 9627 | 29 |
| 90 | 4.1 | 12185 | 32.4 |

Measured $B_{1+rms}$ are those provided by the scanner at the end of heating (dynamic acquisition #90). Energy values of the heating module are computed from the sequence. The last column reports the maximal temperature values measured by the proposed MRI thermometry method.

heating module ranging from 0 to 90°. Maximal $B_{1+rms}$ values were 4.1 µT for a 90° $FA_{HM}$. The maximal temperature as a function of the $FA_{HM}$, $B_{1+rms}$ and energy is displayed in Fig 5, together with the fits. Coefficients resulting from the fits were $\beta_1 = 4.3 \pm 0.1.10^{-3}$°C/°$^2$, $\beta_2 = 2.0 \pm 0.05$°C/(µT)$^2$ and $\beta_3 = 2.3 \pm 0.1~10^{-3}$°C/J, respectively. A strong correspondence was found between experimental data and fits ($R^2 = 0.98$ for each fit).

## Applicability of the method on a pacemaker lead

A local temperature rise up to 6.5°C was observed near the tip of the device (Fig 6A right) for a 90° flip angle. A small (2x2 pixels) hypo intense (less than 20% reduction in intensity) region was observed in the central slice of the magnitude image of the thermometry sequence near the tip of the wire. However, temperature SD measured in these pixels was identical to values measured everywhere else into the gel. Calibration curves are displayed in Fig 6B, with $\beta_1$, $\beta_2$ and $\beta_3$ values of $8.8\pm0.5.10^{-4}$°C/°$^2$ ($R^2 = 0.96$), $0.40\pm0.02$°C/µT$^2$ ($R^2 = 0.97$) and $5.8 \pm0.3.10^{-4}$°C/J ($R^2 = 0.96$), respectively.

## Prediction of temperature rise for three MRI sequences

Fig 7 presents results from an additional experiment performed in a gel to evaluate the proposed model and processing technique. The temperature curves of the calibration data set (Fig

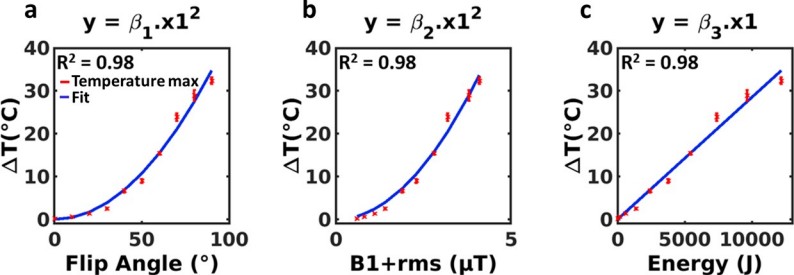

**Fig 5. Dependence of temperature increase in a single pixel on flip angle, $B_{1+rms}$ and deposited energy by the MRI sequence.** Each point corresponds to the mean of the temperature over 5 dynamic acquisitions at the end of the energy deposition (acquisition #90) obtained for $FA_{HM}$ ranging from 0° to 90°. The same pixel was selected for each experiment. Error bars correspond to the $\sigma_T$ over the same 5 dynamic acquisitions. A quadratic curve fit was performed for the two first curves and a linear fit for the last one. Coefficients resulting from the fits were $\beta_1 = 4.3 \pm 0.1.10^{-3}$°C/°$^2$, $\beta_2 = 2.0 \pm 0.05$°C/(µT)$^2$ and $\beta_3 = 2.3 \pm 0.1~10^{-3}$°C/J, respectively.

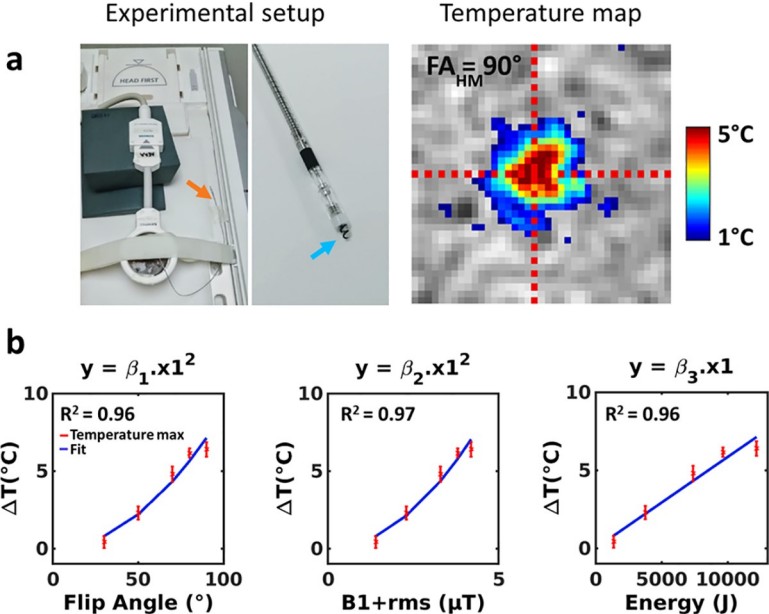

**Fig 6. Application of the proposed method on a pacemaker lead.** (a) Experiments on a MR conditional pacemaker lead inserted into a gel. Left: photographs of the setup showing the position of the gel and lead on the MRI table (orange arrow indicates the position of the extremity of the lead, blue arrow indicated the tip screwed into the myocardium). Right: temperature image at dynamic acquisition #90 for a 90° flip angle of the heating module. (b) Calibration curves obtained for a series of measurements with flip angles of 30°, 50°, 70°, 80° and 90°. Coefficients resulting from a quadratic fit were $\beta_1 = 8.8 \pm 0.5.10^{-4}$ °C/°$^2$, $\beta_2 = 0.4 \pm 0.02$ °C/$(\mu T)^2$ and coefficient resulting from a linearly fit was $\beta_3 = 5.8 \pm 0.3.10^{-4}$ °C/J.

7A) were fit with Eq [1] for each flip angle of the $FA_{HM}$. The resulting $\alpha$ and $\tau$ values derived from these fits are plotted as a function of the corresponding powers in Fig 7B and 7C, together with the results of the polynomial fits. Fig 8A displays the temperature values measured by the optic fiber during a 2D cine, a 2D turbo spin-echo and a 3D gradient echo, emitting 43.59 W during 9 s, 11.09 W during 38 s and 2.52 W during 135 s, respectively. For each sequence, the temperature curves were simulated using Eq [1], after calculating $\alpha$ and $\tau$ from the polynomial fits shown in Fig 7B and 7C. A good correspondence can be observed between experimental and simulated curves, with maximal values of 11.19°C (Cine) 5.62°C (2D TSE) and 2.32°C (3D GRE) for experimental values and predicted values of 11.88°C, 5.85°C and 2.33°C,

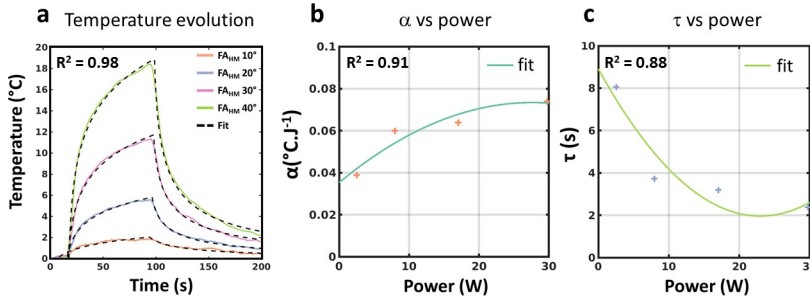

**Fig 7. Temperature data for 4 different $FA_{HM}$.** (a) experimental and fitted curves for $FA_{MH}$ of 10°, 20°, 30° and 40°. (b) Plot of $\alpha$ as a function of power. (c) Plot of $\tau$ as a function of power. In (b) and (c), solid lines represent the result of the polynomial fit with the resulting equations: $\alpha = -4.996 \, P^2 + 2.758 \times 10^{-2} \, P + 3.531 \times 10^{-2}$ and $\tau = 1.316 \times 10^{-2} \, P^2 - 6.049 \times 10^{-1} \, P + 8.912$.

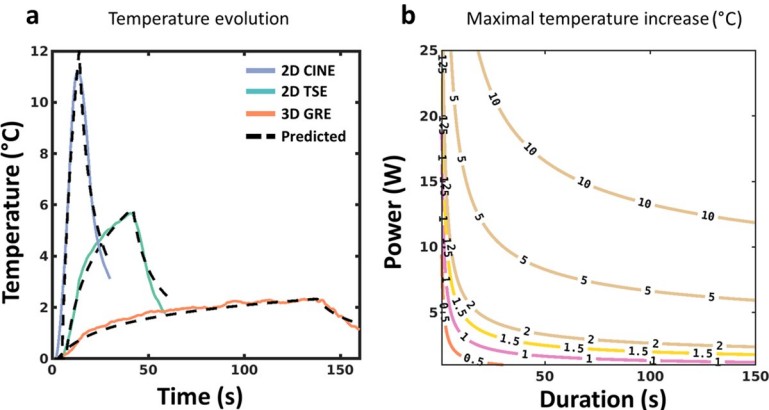

**Fig 8. Prediction of maximal temperature rise for other acquisition sequences.** (a) Predicted (dashed lines) and measured temperature evolution by the optical fiber for the 2D cine (solid blue curve), the 2D TSE (solid green curve) and 3D GRE sequences (solid orange curve). (b) Contour plot of isotherms showing the predicted maximal temperature increase as a function of the duration and power of a MRI acquisition sequence. Polynomial functions displayed in Fig 7 were used in the simulation.

respectively. On the contrary, using the calibration curves (namely coefficient $\beta_3$) from Fig 5C to estimate the maximal temperature increase from the total energy of each sequence leads to 2.78˚C, 2.96˚C and 2.42˚C, respectively. Such an approach is thus irrelevant for predicting maximal temperature rise in the context of an implanted wire.

Fig 8B shows contour plots of the predicted maximal temperature increases for a range of power and duration using the fitting method and for this experimental configuration. Isotherms can then be used to define acceptable exposure conditions for any sequence.

## Discussion

### Sequence implementation

The hybrid sequence proposed in this study interleaves adjustable RF energy deposition with multi-slice EPI acquisitions to provide sufficient spatial (1.6 mm in-plane interpolated to 0.8 mm) and temporal resolution (1s refresh rate with a 3 s temporal footprint in our setting) for real-time visualization of the potential temperature increase in-situ. Similar approach was already proposed in Gensler et al., using T1 measurements to assess temperature evolution near a copper wire inserted into a gel. In plane spatial resolution was 2.3 mm with slice thickness of 5 mm, leading to an elementary voxel size of 26.5 mm$^3$, much larger (factor 4) than in the present study (6.1 mm$^3$). Ehses et al. proposed a PRF-based MR-thermometry method with similar in-plane spatial resolution (1.6 mm) but a slice thickness of 5 mm. Moreover, temporal resolution was 3.9 s per slice, making rapid and multi-slice monitoring of temperature evolution near the wire more difficult than in the present study. Here, the achieved spatial resolution was considered sufficient for observing local temperature hot spots near the wire tip, since the heating region observed in Fig 4A had a dimension ranging from 2.4 to 5.6 mm (full width at half maximum of temperature profile for $FA_{HM} \geq 20$˚). The proposed implementation provides flexibility between the number of slices to acquire volumetric temperature data and energy deposition duty cycle (72% in our experiment). Higher acceleration factors, partial Fourier sampling [36] and/or simultaneous multi-slice techniques [37] may be implemented to increase volume coverage at constant acquisition time.

## Precision of the MRI-thermometry

In our implementation, a temperature uncertainty of ~0.2˚C was obtained on a clinical 1.5T MRI scanner (both on phantom and in vivo in the human brain). Such a precision was better than those previously reported [24, 26] and was considered sufficient in the context of MR safety evaluation of devices, where maximum temperature should not exceed 39˚C for the brain (IEC-60601 and FDA regulation). Although optimal value for PRFS thermometry is achieved when TE equals T2*, the uncertainty in the human brain was found good enough for TE values ranging from 22 ms to 50 ms since they remained below 0.25˚C for at least 75% of the pixels located in the brain irrespective of echo time.

In a previous study, a similar thermometry technique without the heating module showed a good precision (around 1˚C) in vivo in mobile organs such as the heart [29, 30] and the liver at 1.5T [38] including real time motion compensation and correction of the potential temporal drift of the magnet (also implemented in the present work). In the context of monitoring small temperature increases near implanted wires in mobile organs, more sophisticated filtering techniques could be used to further improve the precision of thermometry as proposed by Roujol et al. [39] for example. Despite the proposed implementation creates a latency of 3 s, the risk of reaching a lethal thermal dose within this time scale (according to the CEM43 [40] model) which is unlikely to occur with optimized clinical devices.

## Calibration technique

In the experiments with the copper wire, we observed an important temperature increase when the flip angle of the heating module was higher than 30˚, although $B_{1+rms}$ values provided by the scanner interface remained within regulatory limit (maximal $B_{1+rms}$ of 3.2 μT, as indicated in the fixed-parameter option of IEC 60601–2-33) for most of the experimental conditions ($FA_{HM}$ up to 70˚, see Table 1). As expected from the theory [6], the maximal experimental temperature showed a quadratic variation ($β_1$ and $β_2$ coefficients) with the flip angle and $B_{1+rms}$, together with a linear dependence ($β_3$ coefficient) with the emitted energy. In the experiment with the MR conditional pacemaker lead, we were able to perform identical experiments and obtain different calibration curves, with maximal temperature increase of 6.5˚C.

## Prediction of temperature increase from the model

In the last experiment (Figs 7 and 8), we illustrate that the proposed model can correctly fit the temperature curves of the calibration dataset. From these fits, we show that temperature evolution of other acquisition sequences can be reasonably estimated. An exponential fitting function was proposed in the literature to model temperature increase during the heating phase. However, experimental results reported in Ehses et al. [24] and Gensler et al. [26] did not perfectly fit the MR-temperature curves and diverged from fiber optic measurements using this model. Here, we chose a more physically-realistic model of temperature evolution (derived from a Gaussian-shaped heating source), although this shape is an approximation for RF-induced heating. This model fits both the heating and cooling phases of the temperature curve. Whatever the model, indeed, the two parameters resulting from the fit (here α and τ) are directly linked to tissue absorption and thermal diffusivity, and are thus not expected to vary with the emitted power, at least for moderate temperature increases (i.e. remaining below the lethal thermal dose). However, parameters α and τ (Fig 7B and 7C) derived from the fit of temperature curves (Fig 7A) show a strong variation when varying $FA_{HM}$ from 10˚ to 20˚ and lower changes for higher $FA_{HM}$ (30˚ and 40˚). For these reasons, we introduced a polynomial fitting function in Fig 7B and 7C. Variation of α and τ as a function of the power was attributed to the relative small dimensions of the heating spot for low $FA_{MH}$ values, where partial volume

effect of the thermometry sequence may play a role, although an effort was made to provide high resolution temperature images with the proposed sequence. This semi-empirical model allowed to predict the maximal temperature increase for three other MRI sequences. However, a key point of the calibration step is to avoid creating excessive temperature increase, since protein denaturation can occur when absolute temperature reaches 43˚C (i.e. 6˚C temperature rise above physiological body temperature). In our results, we reached much higher maximal temperature rises when wires were included in the gel. However, in our implementation, we chose a long duration of the heating module (80 s) to validate the acquisition method and the associated processing. Shorter duration of the heating module may be considered to reduce the temperature rise, taking advantage of the 1 s temporal resolution of our thermometry sequence to sample the temperature curve and thus derive $\alpha$ and $\tau$, without inducing excessive temperature increase. Moreover, sampling the flip angles from 0 to 90˚ by 10˚ steps is probably not mandatory since risks are mainly associated with high power deposition, which correspond to large flip angles. This was observed on the pacemaker lead where significant heating was only observed for large flip angles (FA$_{MH}$ of 70˚ and higher in see Fig 6B). Such an optimization of the calibration process was considered out of the scope of the present work, whose objective was to present the acquisition sequence and processing method and to evaluate them under well controlled experimental conditions, as a proof of concept.

The resulting temperature increases may differ in vivo since absorption, thermal diffusivity and perfusion (not present here) are tissue-specific, resulting in different calibration data sets. However, the method is expected to remain valid since perfusion acts as a scaling factor in temperature evolution. Thus, by generating calibration data set at the beginning of the MRI session, it should be possible to determine personalized RF exposure conditions for each patient with an implanted wire. In this objective, real-time MRI-thermometry as proposed here is of central interest to avoid creating excessive temperature rise during the calibration process.

## Study limitations

This study has some limitations. First, PRFS technique is not applicable in fatty tissue. Second, our method may be dependent on size and magnetic susceptibility, limiting its applicability, since local image distortion and signal losses can be particularly severe with echo planar imaging. In our experiments, although the implanted wire was systematically positioned orthogonally to B$_0$, we were able to obtain temperature curves of sufficient quality to successfully process the data. For tissue with long T2* such as in the brain, echo time can be reduced in the presence of an implanted wire to balance the effect of local susceptibility artifacts, while keeping acceptable temperature accuracy (see Fig 2B). Moreover, shortening the echo train duration of the EPI by parallel imaging contributes to reducing susceptibility artifacts. Although EPI suffers from known limitations, this technique was preferred for the aforementioned advantages (rapid and multi slice imaging, high duty cycle), especially given that this technique is available on any scanner and that MRI compatibility of medical devices having implanted wires is under constant improvement by manufacturers [41]. In the present work however, no in vivo data with implanted wires could be produced to assess the method in real conditions, justifying further studies.

## Conclusion

We propose here a practical MRI-based method to monitor the risk of heating during RF deposition by a MRI sequence through direct measurement of local temperature increase. This method may be combined with other MR-based approaches [21, 22] that aim to measure

effective current induced in the device. The proposed method could be used at preliminary stage of the design of new devices with implanted wires to quantify the risk of heating depending on the exposure conditions, using phantoms with tissue-mimicking absorption and thermal diffusivity for example. In patients having devices with implanted wires, this method might be used at the beginning of the MRI session to assess acceptable exposure conditions. This will however require optimization of the calibration process and further in vivo evaluation.

## Author Contributions

**Conceptualization:** Marylène Delcey, Pierre Bour, Valéry Ozenne, Bruno Quesson.

**Funding acquisition:** Valéry Ozenne, Bruno Quesson.

**Investigation:** Marylène Delcey.

**Methodology:** Marylène Delcey, Pierre Bour, Valéry Ozenne, Wadie Ben Hassen, Bruno Quesson.

**Project administration:** Bruno Quesson.

**Supervision:** Bruno Quesson.

**Validation:** Bruno Quesson.

**Writing – original draft:** Marylène Delcey, Bruno Quesson.

**Writing – review & editing:** Marylène Delcey, Pierre Bour, Valéry Ozenne, Bruno Quesson.

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
