## [Decision Letter · Decision Letter 0]

24 Nov 2020

PONE-D-20-32401

A fast MR-thermometry method for quantitative assessment of temperature increase near an implanted wire

PLOS ONE

Dear Dr. Delcey,

Thank you for submitting your manuscript to PLOS ONE. After careful consideration, we feel that it has merit but does not fully meet PLOS ONE’s publication criteria as it currently stands. Therefore, we invite you to submit a revised version of the manuscript that addresses the points raised during the review process.

The manuscript has been reviewed by two reviewers with significant experience in MR thermometry. Both feel that a major revision is necessary to clarify the methods, motivate the application of the technique and potentially expand the scope of work. Please provide a detailed response to all comments made.

We look forward to receiving your revised manuscript.

Kind regards,

Nick Todd, PhD

Academic Editor

PLOS ONE

Journal Requirements:

2. Thank you for including your ethics statement:  "Institution of review board "Commité de protection des personnes îles de France IV", #IRB 0003835

approval number: 2017-A03313-50

written consent ".   

Please amend your current ethics statement to confirm that your named institutional review board or ethics committee specifically approved this study.

"I have read the journal's policy and the authors of this manuscript have the following competing interests: Marylene Delcey and Wadie Ben Hassen are employees of Siemens Healthcare."

We note that one or more of the authors are employed by a commercial company: Siemens Healthcare.

3.1. Please provide an amended Funding Statement declaring this commercial affiliation, as well as a statement regarding the Role of Funders in your study. If the funding organization did not play a role in the study design, data collection and analysis, decision to publish, or preparation of the manuscript and only provided financial support in the form of authors' salaries and/or research materials, please review your statements relating to the author contributions, and ensure you have specifically and accurately indicated the role(s) that these authors had in your study. You can update author roles in the Author Contributions section of the online submission form.

3.2. Please also provide an updated Competing Interests Statement declaring this commercial affiliation along with any other relevant declarations relating to employment, consultancy, patents, products in development, or marketed products, etc.  

Reviewers' comments:

Reviewer's Responses to Questions

**Comments to the Author**

1. Is the manuscript technically sound, and do the data support the conclusions?

Reviewer #1: Yes

Reviewer #2: Partly

2. Has the statistical analysis been performed appropriately and rigorously? 

Reviewer #1: Yes

Reviewer #2: Yes

3. Have the authors made all data underlying the findings in their manuscript fully available?

Reviewer #1: Yes

Reviewer #2: Yes

4. Is the manuscript presented in an intelligible fashion and written in standard English?

Reviewer #1: Yes

Reviewer #2: Yes

5. Review Comments to the Author

Reviewer #1: This work presents a method to assess heating near implanted wires during an MRI scan due to RF

deposition and induced currents. It uses fast measurements of temperature using PRFS MR

Thermometry interleaved with an RF heating module. The method was shown to be precise and

accurate in a variety of ex vivo heating scenarios and an in vivo no-heat scenario. Experiments and

analysis are scientifically sound. I appreciated the inclusion of the limitations and comparison to existing

methods in the discussion. Clarification of the scope, methods, and figures are needed.

After these adjustments, I believe the article is suitable for publication.

Specific comments:

1) The title implies the work is for measurement near implanted wires, but the intro (page 5, line

99), methods (page 8 line 172), and others refer to "implants” or “devices." This terminology

should be clarified throughout, as no evidence is presented to discuss how the work would be

applicable to other types of implanted devices.

2) The use case for the work should be made clear. How do you see the method being used in

practice? Is the method intended to be used during real-time monitoring, as a one-time

phantom calibration that exists in look-up-table form for each wire type, or something else?

How would this be used safely at the beginning of each MRI session (page 19, line 384) without

applying heating to the patient? Clarifying this in the abstract, introduction, and discussion will

help clearly define the impact of the work.

3) How much artifact was seen around the tips of the lead wires tested? Did it interfere with

maximum heat voxel selection/voxel near thermocouple selection? How would the method

perform if the lead wire were oriented differently with respect to B0, thus potentially having a

larger artifact?

4) Is the low-pass filtering necessary for generating the calibration curves, or would the original

data be precise enough? Would the filter latency (page 12, line 246) be an issue for clinical

implementation?

5) Figure 1 could be expanded to include which dynamics were averaged together. Add a label the

thermometry module.

6) Add titles to all sub-parts of Figures 2 and 3, and 6a.

7) Scale bar in Fig 2A blends in, consider a different color and moving outside the phantom into the

black area and adding the value to the image.

8) Figures 3c, 4b, 5, 6b need plot legends.

9) Page 6 line 127: How long was each dynamic (with and without interleaving the heating

module)?

10) Page 9 line 175: How was the voxel of interest chosen?

11) Page 9 line 179: add scan parameters to statistics section

12) Page 9 line 189: is this FA for the thermometry imaging module or the RF heating module?

13) Page 10 line 199: Was there any appreciable cooling during the 5 dynamics that were averaged?

Since this depends on # of slices, how many slices were used in each of the experiments?

14) Page 10 line 202: Explain how energy emitted was computed.

15) Page 16 lines 307-314 should be in the methods section.

16) Page 18 line 373: What is the regulatory limit?

Reviewer #2: This manuscript describes a method to run an MRI pulse sequence with an added “Heating module” to induce heating in implanted wires. The pulse sequence is a single-shot EPI pulse sequence for PRF MR thermometry, previously described in multiple papers by the same group. If the sequence is run multiple times with different parameters for the “Heating module”, the flip angle, B1+, and Energy can be plotted as functions of the temperature increase measured (with PRF MR thermometry) at the tip of a wire/lead. This creates what the authors call calibration curves, and which they claim can then be used to be sure temperature increases stay below regulatory limits even when other pulse sequences are used. Experiments were performed in agar gel phantoms to investigate accuracy (as compared to fiber optic temperature measurements) and precision of the MR thermometry, and in one healthy volunteer to investigate precision in vivo in brain.

Over all the approach is interesting and probably worth investigating. It is however not clearly described in the manuscript how the authors envision the approach being used, and the reader has to “read between the lines” to really understand the point of the “calibration curves”. This should be made clearer and described more straight forward in the introduction.

Secondly, it’s not clear why the authors went through the trouble of doing all these experiments and stopped short of actually evaluating the method for its intended purpose. The whole point of getting the calibration curves are so you can predict how much heating other (more clinically relevant) pulse sequence will induce. So, when doing the phantom experiment with the fiber optic probe, why didn’t the authors derive the calibration curves and then used them to predict how much heating a set of clinically relevant pulse sequences would induce, and then compare to what the probe actually measured? Without this experiment, the paper will be of very limited impact as it is not clear if the described method will actually work as intended. In my opinion this experiment (at the very least in phantom or maybe better in ex vivo tissue, but ideally in vivo in an animal model) must be included before the manuscript can be published. When doing this experiment the maximum temperature rise when getting the calibration curves should ideally be kept below 6 °C as this is generally when thermal dose starts to accumulate in vivo (i.e., at 43 °C assuming 37 °C starting temperature). Without this experiment the authors can probably not make statements/claims such as “Calibration curves derived from temperature measurements under different RF exposure levels were fit to predict temperature increase for any MR-acquisition sequence”.

Lastly, the orientation of leads/wires inside the bore can affect how much heating and artifacts are created, and it is not clear how well the single-shot EPI sequence handles this. This is another straight forward experiment to perform that would improve the readers excitement about the paper.

Minor comments:

Abstract

“...compared to invasive fiber-optic measurements to assess precision…“ this would assess accuracy and not precision?

“In gel, as well as in the human brain, temperature measurements within ± 0.2 °C

certainty” Please reformulate this. I assume this is from the SD through time, so maybe say something like “the precision of the measurement was 0.2 °C…”. Also mention that this is after temporal filtering.

Ln 117: The Introduction discusses the importance of a large enough FOV – why was such a small FOV (only 12 cm) chosen? That’s not practical for anything but maybe imaging extremities – certainly too small for head and body imaging.

Ln 118: How many slices were interleaved in the 1000 ms TR?

Ln 124: Change KHz to kHz

Ln 125: TR is used above – define when it’s first used (and define other parameters above, too).

Ln 136: What algorithm was used for EPI ghost correction?

Ln 138: Please change “pixel size” to “pixel spacing” (the size doesn’t change with zero filling, but spacing does)

Ln 152-154: This sentence is not clear. Did it take 6 minutes before you could run the sequence again? Why was that?

Ln 171-172: When removing probes from gel phantoms an air-filled “track” is often left behind, resulting in susceptibility artifacts – did you observe this? Or did you use a new/separate phantom for this experiment?

Ln 174: Doing 10 heatings in a single location can seem like quite a lot – did you somehow check/control that the heatings were repeatable, by, e.g., repeating the same heating at the beginning, middle and end?

Ln 211-212: Please use same number of significant figures for all numbers

Ln 252: Suggest change “given” to “measured”

Ln 306: Most (all?) of this paragraph seems to belong better in the Materials section. This whole experiment wasn’t mention in the Materials at all.

Ln 339: Again, interpolation doesn’t change the size of the voxel, just the spacing. So, the voxel size is 6.3 mm3 both before and after zero filled interpolation.

Ln 348-350: Changing the TR will change the duty cycle no matter if the heating is fast or slow, so why do you say “since the temperature evolution is relatively slow”? Even if it was fast the duty cycle would change with changing TR? Or do you mean something else?

Ln 354-356: This was, however, after temporal filtering. Please add if references 24 and 26 also used temporal filtering. If they didn’t, how did your unfiltered values compare to theirs?

Ln 386-388: “In this objective, real-time MRI-thermometry as proposed here is of central interest to avoid creating excessive temperature rise during the calibration process.” Well, yes, but you need the MR thermometry to create the curves in the first place.

Ln 396: Do you mean Figure 2b?

References

Please check all references. They seem to contain months (and other things?) in French rather than English, etc.

Figures

The Figures are overall fairly low quality (at least in the provided pdf), so it’s hard to see any details. Please include higher resolution figures.

6. PLOS authors have the option to publish the peer review history of their article (what does this mean?). If published, this will include your full peer review and any attached files.

Reviewer #1: No

Reviewer #2: No

---

## [Author Response · Author response to Decision Letter 0]

5 Feb 2021

Response to Reviewers

We thank both reviewers for their overall positive evaluation of our work and suggestions for improving the manuscript. We understand that the main criticisms were:

 Clarify how calibration curves can be used to actually evaluate the temperature rise of any MRI sequence

 How can this technique be used practically

In our initial manuscript, we wrote that calibration curves can be used to predict maximal temperature rise for any other MRI sequence. The idea of these calibration curves was to provide quantitative estimate of maximal temperature rise near the tip of the wire as a function of the total energy (through the fits of maximal �T vs flip angle, B1+rms or Energy, providing �1, �2 and �3 coefficients). From these fits, one would then be able to estimate the temperature rise of any other MRI sequence based on energy of the sequence only. This idea was driven by the current safety approach in MRI, where SAR and B1+rms are commonly accepted quantities to assess safety, both being time-averaged values. 

When reading comments from the reviewers (and particularly those from reviewer #2), we realized that this idea is in fact incorrect in the context of an implanted wire, since temperature rise effectively depends on the emitted power and duration (thus is energy dependent) of a given MRI sequence, but that the most relevant parameter is the power since it is linked to the temperature rise. As an example, 100 W applied during 10s will produce a much higher heating than 1 W applied during 1000 s, although energy is identical. We thus worked to solve this issue and to improve the manuscript by providing a method that takes into account the power emitted by the sequence instead of its cumulative energy or B1+rms.

We introduced in the revised manuscript a temperature model derived from previously published work on high intensity focused ultrasound. Using this approach, we can exploit the temperature curves obtained from the SAR module sequence in a different manner. First, we approximated the nearly punctual heating source (tip of the wire) as a Gaussian shape. This assumption offers the advantage of allowing to use an analytical function of temperature evolution at the hottest point, already referenced in the literature for HIFU heating (where a Gaussian function is considered as a realistic model). 

Then, we fit temperature evolution for each flip angle of the SAR module with this function to derive two parameters (α and �) linked to tissue characteristics (absorption and thermal diffusivity). 

For a MRI sequence depositing a power P0 over a duration D, one can then simulate temperature evolution and predict maximal temperature rise, which is the relevant parameter for safety consideration regarding IEC standards. This semi-empirical model is presented together with an additional data set, comparing temperature rises measured by an optical fiber with those predicted by the model for three MRI sequences having different duration and emitted power. We show that the method is accurate for predicting the maximal temperature rise within 5% error. 

We believe these amendments clarifies the focus of the paper and provide more evidence of the interest and potential of the proposed method. We also discuss the potential operating modes in perspective of future use. 

We unfortunately could not implement an in vivo experiments to evaluate this method due to several practical constraints. However, we think this paper is a proof of concept and we hope this new manuscript will receive a positive evaluation.

We again thank both reviewers for their comments that helped us to improve the manuscript and avoid publishing incorrect conclusions.

 

Reviewer #1: This work presents a method to assess heating near implanted wires during an MRI scan due to RF deposition and induced currents. It uses fast measurements of temperature using PRFS MR Thermometry interleaved with an RF heating module. The method was shown to be precise and accurate in a variety of ex vivo heating scenarios and an in vivo no-heat scenario. Experiments and analysis are scientifically sound. I appreciated the inclusion of the limitations and comparison to existing methods in the discussion. Clarification of the scope, methods, and figures are needed. After these adjustments, I believe the article is suitable for publication.

Introduction was amended to clarify the scope. The method section now includes a processing technique to predict temperature evolution for other MRI sequence from temperature data obtained during the calibration phase. New results have been added to illustrate this technique using 3 different MRI sequences with different power and duration. The discussion has also been amended accordingly.

Specific comments:

R1.1) The title implies the work is for measurement near implanted wires, but the intro (page 5, line 99), methods (page 8 line 172), and others refer to "implants” or “devices." This terminology should be clarified throughout, as no evidence is presented to discuss how the work would be applicable to other types of implanted devices.

We modified the text to remove any ambiguity since the method is effectively intended to assess temperature increase near an implanted wire.

R1.2) The use case for the work should be made clear. How do you see the method being used in practice? Is the method intended to be used during real-time monitoring, as a one-time

phantom calibration that exists in look-up-table form for each wire type, or something else?

How would this be used safely at the beginning of each MRI session (page 19, line 384) without applying heating to the patient? Clarifying this in the abstract, introduction, and discussion will help clearly define the impact of the work.

We addressed this question in the introduction, discussion and conclusion of the revised manuscript. Objectives of the method is three fold: 1) to present a real time and precise temperature monitoring technique with adequate spatial and temporal resolutions for the targeted application, 2) to create a calibration data set by exploiting this sequence using interleaved heating module with frequent (1s update rate) temperature estimates, and 3) propose a model for predicting the maximal temperature increase for other MRI acquisition sequences from this calibration dataset, without requiring sophisticated modelling or using surrogates of temperature measurements. 

This manuscript aims at presenting the proof of concept and potential optimizations are proposed in the discussion section to avoid inducing excessive heating (as pointed out by reviewer #2 also) in the patient.

We propose typical use cases in the conclusion of the manuscript.

R1.3) How much artifact was seen around the tips of the lead wires tested? Did it interfere with

maximum heat voxel selection/voxel near thermocouple selection? How would the method

perform if the lead wire were oriented differently with respect to B0, thus potentially having a

larger artifact?

We are aware that this could be a limitation of the imaging sequence. However, in each experiment presented in this manuscript, the implanted wire was systematically positioned vertically into the gel, thus orthogonally to B0. This is now indicated in the material and method section for each experiment. No visible artifact could be identified in the thermometry slice crossing the tip of the implanted wire when no fiber-optic probe was present. For the pacemaker lead a small reduction in signal intensity was observed in 4 pixels but without impacting significantly the thermometry precision (also indicated into the revised manuscript). 

R1.4) Is the low-pass filtering necessary for generating the calibration curves, or would the original data be precise enough? Would the filter latency (page 12, line 246) be an issue for clinical implementation?

The low-pass filter was introduced to reduce noise on temperature images. Whether this is necessary or not is directly related to the standard deviation of the thermometry. In our implementation at 1.5T, with the selected coils on these gels with the selected spatial resolution and echo time, we decided to include it to reach good temperature precision. The latency induced by this filter is acceptable (3 seconds) since in case an important temperature increase is observed, the sequence can be immediately stopped, without creating risks for the patient. In 3 s, a temperature of 55°C (+18°C above body temperature) is necessary to reach the lethal thermal dose (taking the CEM43 metric). Other types of filters may be included in the thermometry pipeline such as Kalman filters (as already stated in the discussion section of the original manuscript). However, we do not consider the current implementation is an issue for clinical application. 

A sentence has been added in the discussion section:

“However, despite the proposed implementation creates a latency of 3 s, a temperature increase of 18°C is necessary to reach a lethal thermal dose within this time scale (according to the CEM43 model) which is unlikely to occur with optimized clinical devices.” 

R1.5) Figure 1 could be expanded to include which dynamics were averaged together. Add a label the thermometry module.

Done

R1.6) Add titles to all sub-parts of Figures 2 and 3, and 6a.

Done

R1.7) Scale bar in Fig 2A blends in, consider a different color and moving outside the phantom into the black area and adding the value to the image.

We changed the bar color and location to avoid such a blend. 

R1.8) Figures 3c, 4b, 5, 6b need plot legends.

Done

R1.9) Page 6 line 127: How long was each dynamic (with and without interleaving the heating

module)?

Each dynamic was 1 sec long and maintained constant during the acquisition. When heating module is played, the remaining “empty” delay before the following acquisition of slices is filled with pulses (see figure 1 and M&M section for details) 

“TR/TE = 1000/18 ms” was already indicated in the original manuscript

We kept the original text unchanged.

R1.10) Page 9 line 175: How was the voxel of interest chosen?

The sentence was amended as follow:

“For each acquisition, the temperature evolution in the same pixel was analyzed, selecting the pixel with the maximal temperature increase at the end of the energy deposition (acquisition #90) from the temperature data corresponding to the largest FAHM”

R1.11) Page 9 line 179: add scan parameters to statistics section

The following sentence was added. ” The same scan parameters as described in section MRI-thermometry sequence were used”

R1.12) Page 9 line 189: is this FA for the thermometry imaging module or the RF heating module?

This FA is the flip angle of the thermometry sequence, since here the heating module emitted no energy (FAHM=0°) 

R1.13) Page 10 line 199: Was there any appreciable cooling during the 5 dynamics that were averaged? Since this depends on # of slices, how many slices were used in each of the experiments?

No cooling was observed during the 5 successive acquisitions (see temperature curves in Fig 4b). Three slices were used in each of the experiments (already indicated in the original manuscript). The text in the manuscript was not modified, but we added a mark in Fig 4b (FAHM=40°) showing the temporal window corresponding to this averaging.

This will help the reader to see that temperature can be considered nearly constant over this temporal window.

R1.14) Page 10 line 202: Explain how energy emitted was computed.

The sentence was modified as follow:

“ The µT ± σT temperature values were plotted as a function of the flip angle, B1+rms and energy emitted (i.e. sum of the energies of each individual RF pulse, including pulses for imaging and pulses of the heating module) “

R1.15) Page 16 lines 307-314 should be in the methods section.

We moved this section to Methods section.

R1.16) Page 18 line 373: What is the regulatory limit?

In the latest version of IEC 60601–2‐33, the so‐called fixed‐parameter option (FPO) was introduced for 1.5T systems (FPO:B), which specifically addresses the scanning of implant carriers and fixed limit value of B_(1,rms)^+ = 3.2 µT. In Table 1, this correspond to FAHM of 70° .

The sentence was amended as follow:

“In the experiments with the copper wire, we observed an important temperature increase when the flip angle of the heating module was higher than 30°, although B1+rms values provided by the scanner interface remained within regulatory limit (maximal B1+rms of 3.2 µT, as indicated in the fixed‐parameter option of IEC 60601 –2‐33) for most of the experimental conditions (FAHM up to 70 °, see Table 1).”

Reviewer #2: This manuscript describes a method to run an MRI pulse sequence with an added “Heating module” to induce heating in implanted wires. The pulse sequence is a single-shot EPI pulse sequence for PRF MR thermometry, previously described in multiple papers by the same group. If the sequence is run multiple times with different parameters for the “Heating module”, the flip angle, B1+, and Energy can be plotted as functions of the temperature increase measured (with PRF MR thermometry) at the tip of a wire/lead. This creates what the authors call calibration curves, and which they claim can then be used to be sure temperature increases stay below regulatory limits even when other pulse sequences are used. Experiments were performed in agar gel phantoms to investigate accuracy (as compared to fiber optic temperature measurements) and precision of the MR thermometry, and in one healthy volunteer to investigate precision in vivo in brain. 

R2.1 Over all the approach is interesting and probably worth investigating. It is however not clearly described in the manuscript how the authors envision the approach being used, and the reader has to “read between the lines” to really understand the point of the “calibration curves”. This should be made clearer and described more straightforward in the introduction.

We agree that this was unclear in the original manuscript. We have thoroughly modified the manuscript to explain this in the introduction, added a section in material and methods, include new results and discuss them in the revised manuscript.

We hope these modifications bring clarity and a more precise focus to the manuscript. 

R 2.2 Secondly, it’s not clear why the authors went through the trouble of doing all these experiments and stopped short of actually evaluating the method for its intended purpose. The whole point of getting the calibration curves are so you can predict how much heating other (more clinically relevant) pulse sequence will induce. So, when doing the phantom experiment with the fiber optic probe, why didn’t the authors derive the calibration curves and then used them to predict how much heating a set of clinically relevant pulse sequences would induce, and then compare to what the probe actually measured? Without this experiment, the paper will be of very limited impact as it is not clear if the described method will actually work as intended. In my opinion this experiment (at the very least in phantom or maybe better in ex vivo tissue, but ideally in vivo in an animal model) must be included before the manuscript can be published. 

Thank you very much for this criticism of the original manuscript. We have amended the method to introduce a heating model and propose a method to compute temperature evolution of other sequences from calibration data set using our proposed MR-thermometry sequence. We included an additional experiment to evaluate this method and discuss these points. Unfortunately, we could not organize an in vivo experiment for various practical reasons. This is clearly stated in the revised manuscript and we believe this new version brings enough new material to be considered for publication as a proof of concept. In the discussion, we propose potential improvements of the method and conclude with potential use-case scenarios. 

 We hope all these improvements will convince the reviewer.

R 2.3 When doing this experiment the maximum temperature rise when getting the calibration curves should ideally be kept below 6 °C as this is generally when thermal dose starts to accumulate in vivo (i.e., at 43 °C assuming 37 °C starting temperature). Without this experiment the authors can probably not make statements/claims such as “Calibration curves derived from temperature measurements under different RF exposure levels were fit to predict temperature increase for any MR-acquisition sequence”.

We totally agree with the reviewer. In the revised version of the manuscript we added a paragraph to the discussion section to address this comment.

R2.4 Lastly, the orientation of leads/wires inside the bore can affect how much heating and artifacts are created, and it is not clear how well the single-shot EPI sequence handles this. This is another straight forward experiment to perform that would improve the readers excitement about the paper.

Each experiment presented here was performed with the wire tip positioned vertically into the gel (thus perpendicular to B0). This was thus the worst case configurations. Please see answer to R1.3 for more details.

Minor comments:

R2.5 Abstract

“...compared to invasive fiber-optic measurements to assess precision…“ this would assess accuracy and not precision?

We corrected this in the text.

R2.6 “In gel, as well as in the human brain, temperature measurements within ± 0.2 °C

certainty” Please reformulate this. I assume this is from the SD through time, so maybe say something like “the precision of the measurement was 0.2 °C…”. Also mention that this is after temporal filtering.

We reformulated the sentence as suggested

R2.7 Ln 117: The Introduction discusses the importance of a large enough FOV – why was such a small FOV (only 12 cm) chosen? That’s not practical for anything but maybe imaging extremities – certainly too small for head and body imaging.

We initially chose a small FOV to concentrate on only a local heating at the wire and pacemaker lead tips. However, as depicted in the human volunteer, it is possible to adjust the FOV depending on the region of interest and thus, acquiring with a larger FOV. The meaning of the sequence was that FOV must be large enough to cover regions around the implanted wire, while being rapid, precise and spatially resolved.

The sentence was modified as follow: “In the context of measuring small temperature changes near devices, MRI-thermometry should be rapid with a sufficiently large spatial coverage around the implanted wire and provide a spatial resolution in the range of a few millimeters.”

R2.8 Ln 118: How many slices were interleaved in the 1000 ms TR?

3 slices were acquired every second. This was already indicated in the original text.

R2.9 Ln 124: Change KHz to kHz

Corrected

R2.10 Ln 125: TR is used above – define when it’s first used (and define other parameters above, too).

Corrected

R2.11 Ln 136: What algorithm was used for EPI ghost correction?

Reconstruction pipeline included EPI ghost correction using three central line of k-space. We refer the reader to the paper by Ozenne et al for implementation details (reference #29 in the manuscript). 

R 2.12 Ln 138: Please change “pixel size” to “pixel spacing” (the size doesn’t change with zero filling, but spacing does)

We proceeded to change

R2.13 Ln 152-154: This sentence is not clear. Did it take 6 minutes before you could run the sequence again? Why was that?

On the MR scanner, it takes 6 minutes to reset the B1+rms value to zero. As we wanted to consider the B1+rms emitted by one sequence only we had to wait this delay. Acquiring before this delay would have result in a mix of B1+rms emitted by several sequences. Moreover, this 6 minute delay ensured proper cooling of the gel before the following experiment with a different FAHM.

The text was amended accordingly.

R 2.14 Ln 171-172: When removing probes from gel phantoms an air-filled “track” is often left behind, resulting in susceptibility artifacts – did you observe this? Or did you use a new/separate phantom for this experiment?

A new phantom was use for each separate experiment to avoid the mentioned issue of susceptibility artifacts induced by air-filled track of previous experiments. This mentioned in the revised manuscript.

R2.15 Ln 174: Doing 10 heatings in a single location can seem like quite a lot – did you somehow check/control that the heatings were repeatable, by, e.g., repeating the same heating at the beginning, middle and end?

After the last heating was performed (largest FAHM value), a delay was observed for cooling down. Then another sequence with FAHM of 30° was acquired again to ensure same value of maximal temperature increase was obtained as for the previous experiment performed with identical FAHM. This was the case, demonstrating repeatability of the heating.

The text was not modified, since we believe this does not bring major added value to the manuscript.

R2.16 Ln 211-212: Please use same number of significant figures for all numbers

This was corrected in the text.

R2.17 Ln 252: Suggest change “given” to “measured”

Done

R2.18 Ln 306: Most (all?) of this paragraph seems to belong better in the Materials section. This whole experiment wasn’t mention in the Materials at all.

We moved this paragraph in the method section, as suggested.

R2.19 Ln 339: Again, interpolation doesn’t change the size of the voxel, just the spacing. So, the voxel size is 6.3 mm3 both before and after zero filled interpolation.

We removed “before interpolation” in the text.

R2.20 Ln 348-350: Changing the TR will change the duty cycle no matter if the heating is fast or slow, so why do you say “since the temperature evolution is relatively slow”? Even if it was fast the duty cycle would change with changing TR? Or do you mean something else?

Of course, changing the TR results in a different duty cycle. This sentence suggested that considering the relative slow evolution of the temperature, the update time of temperature measurement could be increased (eg ever 2 seconds). This increase in TR might be invested in acquiring more EPI slices to increase the spatial coverage of the temperature measurement, if desired. However, this sentence does not appear essential and we prefer to remove it if it is unclear. 

R2.21 Ln 354-356: This was, however, after temporal filtering. Please add if references 24 and 26 also used temporal filtering. If they didn’t, how did your unfiltered values compare to theirs?

Comparing temperature SD results with those from Refs 24 and 26 is not straightforward since it depends on several parameters such as the SNR, which is linked to the voxel size and receiver coil performance. Ehses et al used a PRF thermometry technique with a larger voxel size of 1.56x1.56x5mm (12.168 mm3 vs 6.1 mm3 in our study) with an update time of 3.9 s (versus 1 s in our implementation). They spatially averaged temperature data in 2 adjacent pixels that resulted in 0.5°C standard deviation. Gensler used a T1-based thermometry and reported a temperature standard deviation of 1.37°C. However, their update time was 6.4 s for an elementary voxel size of 26.5 mm3.

In both cases, a precise comparison of their achievements with the standard deviation reported with our method appears hardly feasible, except using the raw values provided in their studies (0.5°C and 1.37°C) and compare them to our 0.2°C after temporal filtering, without taking into account the different voxel dimensions and update times. 

Thus, we prefer to keep the original text unchanged.

R2.22 Ln 386-388: “In this objective, real-time MRI-thermometry as proposed here is of central interest to avoid creating excessive temperature rise during the calibration process.” Well, yes, but you need the MR thermometry to create the curves in the first place.

In this study, we first propose a fast MR-thermometry method. Then, this is used to create a calibration data set from which one can derive maximal temperature increase using a model to fit temperature curves of the calibration data set. We report an example of our method using a medical device where small heating was observed (<6°C). 

As pointed out in the discussion section, the model allows to fit temperature data during heating and cooling. Thus optimization of the calibration process can be envisioned, with the idea of reducing the duration of the heating module (here 80 s). This paper is thus a proof of concept of the method with a lot of potential improvement in perspective of clinical use and will require further studies, including in vivo application.

Please see the discussion section that covers these points.

R2.23 Of course, but having thermometry does not mean that you need to heat excessively. The use of thermometry in real-time avoids reaching excessive temperature (eg no larger than 43°C).

This is now discussed in the revised manuscript.

R2.24 Ln 396: Do you mean Figure 2b?

Yes, thank you for having noticed. 

R2.25 References 

Please check all references. They seem to contain months (and other things?) in French rather than English, etc.

Corrected

R2.26 Figures

The Figures are overall fairly low quality (at least in the provided pdf), so it’s hard to see any details. Please include higher resolution figures.

This is due to the conversion process.

---

## [Decision Letter · Decision Letter 1]

17 Mar 2021

PONE-D-20-32401R1

A fast MR-thermometry method for quantitative assessment of temperature increase near an implanted wire

PLOS ONE

Dear Dr. DELCEY,

Thank you for submitting your manuscript to PLOS ONE. After careful consideration, we feel that it has merit but does not fully meet PLOS ONE’s publication criteria as it currently stands. Therefore, we invite you to submit a revised version of the manuscript that addresses the points raised during the review process.

Both reviewers feel that the manuscript is much improved due to the revisions made and that their major concerns have all been addressed. They also both had a few minor points that would be worth addressing before publication, some about the new material that was added. Please include a formal response to the points raised and make revisions as necessary. If all points are addressed, this round of revision can be taken care of at the editorial level without having to go back out to the reviewers.

We look forward to receiving your revised manuscript.

Kind regards,

Nick Todd, PhD

Academic Editor

PLOS ONE

Journal Requirements:

Reviewers' comments:

Reviewer's Responses to Questions

**Comments to the Author**

1. If the authors have adequately addressed your comments raised in a previous round of review and you feel that this manuscript is now acceptable for publication, you may indicate that here to bypass the “Comments to the Author” section, enter your conflict of interest statement in the “Confidential to Editor” section, and submit your "Accept" recommendation.

Reviewer #1: (No Response)

Reviewer #2: All comments have been addressed

2. Is the manuscript technically sound, and do the data support the conclusions?

Reviewer #1: Yes

Reviewer #2: Yes

3. Has the statistical analysis been performed appropriately and rigorously? 

Reviewer #1: N/A

Reviewer #2: Yes

4. Have the authors made all data underlying the findings in their manuscript fully available?

Reviewer #1: Yes

Reviewer #2: Yes

5. Is the manuscript presented in an intelligible fashion and written in standard English?

Reviewer #1: Yes

Reviewer #2: Yes

6. Review Comments to the Author

Reviewer #1: The manuscript has been greatly improved this round. All previous comments have been addressed. A new set of analysis and figures have been added to address previous concerns around how the work is to be used/calibration curves. These were effective additions. I also appreciate the text that was added to the conclusion discussing how the work can be used. Given the substantial additions to the text and figures, all comments below relate to the added work. Once addressed I believe the article is suitable for publication.

Figure 7 (and associated results/methods text) - Report the goodness of fit. Add text justifying why a second order polynomial should be used. Based on the figure I don't think there is enough data to justify a second order polynomial. Add subfigure labels a and b to the figure itself.

Figure 8 - Add units to the temperature isotherm lines (or indicate in caption). Add subfigure labels a and b to the figure itself.

Page 9 ~line 200 and Page 12 ~line 260 (in tracked changes version) - indicate wire direction relative to scanner B0, not relative to the gel.

Page 12 (in tracked changes version) - Make sure all symbols used in eqn 1 are explicitly defined (I couldn't find T, t, alpha, and tau)

Page 23 line 504 (in tracked changes version) - clarify whether the alpha and tau were allowed to vary in the data shown in the figures. I'm not convinced they should be allowed to vary unless you're seeing temperature rises above coagulations thresholds.

Reviewer #2: Thank you for answering my previous questions and updating the manuscript, which now is clearer and easier to follow. I just have a few minor suggestions on this latest version.

Abstract, Purpose; Suggest adding that you use scans from one sequence to predict heating for other sequences.

Ln 51: Is 0.5 °C maximum error, RMSE etc.? Please clarify.

Ln 204-205: Please reformulate “The tip of the lead screwed into the myocardium was inserted…”

Ln 225-227: What was the TR for these scans? Was it the same for all TEs (if not, compare the precision later on gets challenging)?

Ln 249-250: Does the Gaussian have the same width in all three dimensions (what was it?)? Or is it elongated like most focused ultrasound focal spots?

Ln 350: This reviewer can’t seem to find any blue line (showing “baseline”)? Also, consider calling it “background” rather than “baseline”.

Ln 408-410: It’s a bit unclear what you mean with this sentence - simply that you can use the plot in 5c to estimate the maximum temperature rise?

Ln 436: “(1 s refresh rate in our setting)” please add “with a 5 s temporal foot print”. This is important as you had to shift the temporal curve to align it with the probe measurements. Hence, it is not optimally suited for real-time applications (similar to using a sliding window reconstruction for undersampled k-space data).

Ln 448: Change “FA ≥ 20°” to “FAHM ≥ 20°”, right?

Ln 472: This doesn’t quite tell the full story, right? This assumes a “step function” going straight to 18 °C and holding it there for 3 s. In reality you will start accumulating dose as soon as your increase is 6 °C so with the fairly slow heatings shown in Fig 4 you’ll have a substantial dose before getting to 18 °C.

Figure 8: Both the CINE and TSE predict the heating and start of cooling pretty well, but starts to deviate substantially at the end of the cooling period – can the authors speculate why this is?

7. PLOS authors have the option to publish the peer review history of their article (what does this mean?). If published, this will include your full peer review and any attached files.

Reviewer #1: No

Reviewer #2: No

---

## [Author Response · Author response to Decision Letter 1]

7 Apr 2021

Answer to comments

We thank both reviewers for their positive evaluation of the revised manuscript.

Please find below our point-by-point response to their comments.

Reviewer #1: The manuscript has been greatly improved this round. All previous comments have been addressed. A new set of analysis and figures have been added to address previous concerns around how the work is to be used/calibration curves. These were effective additions. I also appreciate the text that was added to the conclusion discussing how the work can be used. Given the substantial additions to the text and figures, all comments below relate to the added work. Once addressed I believe the article is suitable for publication.

R1.1: Figure 7 (and associated results/methods text) – 

Report the goodness of fit: We reported the goodness of fit within the figure. 

Add text justifying why a second order polynomial should be used. Based on the figure I don't think there is enough data to justify a second order polynomial: 

Please see answer to comment R1.5 below

Add subfigure labels a and b to the figure itself: done

R1.2: Figure 8 – 

Add units to the temperature isotherm lines (or indicate in caption): we added the unit (°C) in the subfigure label for fig b 

Add subfigure labels a and b to the figure itself: 

Done

R1.3: Page 9 ~line 200 and Page 12 ~line 260 (in tracked changes version) - indicate wire direction relative to scanner B0, not relative to the gel.

Done at both places.

R1.4: Page 12 (in tracked changes version) - Make sure all symbols used in eqn 1 are explicitly defined (I couldn't find T, t, alpha, and tau)

This is now detailed. There were also typo errors in the equation that are now corrected.

R1.5: Page 23 line 504 (in tracked changes version) - clarify whether the alpha and tau were allowed to vary in the data shown in the figures. I'm not convinced they should be allowed to vary unless you're seeing temperature rises above coagulations thresholds.

Alpha and tau were allowed to vary in the data shown in the figures. In the discussion section we already mentioned this explicitly and discussed that point:

“the two parameters resulting from the fit (here � and �) are directly linked to tissue absorption and thermal diffusivity, and are thus not expected to vary with the emitted power. However, allowing � and � to vary in our processing improved the fitting quality on each temperature curve of the calibration dataset. This was attributed to the relative small dimensions of the heating spot for low FAHM values, where partial volume effect of the thermometry sequence may play a role, although an effort was made to provide high resolution temperature images with the proposed sequence. This semi-empirical model allowed to predict the maximal temperature increase for three other MRI sequences.”

Of course, alpha and tau may vary with coagulation but this is not the use case of the method since the plan is to create a moderate heating and use temperature curves to predict the maximal temperature rise for any other sequence. 

Moreover, if you compare the values of alpha and tau derived from the fits for the two first experiments (FAHM =10° and 20°) in Fig 7b and 7c, they appear different although there is no expected change in gel composition with temperature increases below 2°C (FAHM=10°) and 6°C (FAHM = 20°) starting from room temperature (~20°C). Thus, we attributed these changes to partial volume effect of the thermometry sequence that is more pronounced for very low temperature increase, despite we tried to use a high resolution thermometry sequence. 

The text was modified as follow:

“Whatever the model, indeed, the two parameters resulting from the fit (here ��and �) are directly linked to tissue absorption and thermal diffusivity, and are thus not expected to vary with the emitted power, at least for moderate temperature increases (i.e. remaining below the lethal thermal dose). However, parameters � and � (Fig 7b and 7c) derived from the fit of temperature curves (Fig 7a) show a strong variation when varying FAHM from 10° to 20° and lower changes for higher FAHM (30° and 40°). For these reasons, we introduced a polynomial fitting function in Fig. 7b and 7c. Variation of � and � as a function of the power was attributed to the relative small dimensions…”

The following sentence was removed:

“However, allowing � and � to vary in our processing improved the fitting quality on each temperature curve of the calibration dataset.”

 

Reviewer #2: Thank you for answering my previous questions and updating the manuscript, which now is clearer and easier to follow. I just have a few minor suggestions on this latest version.

R2.1: Abstract, Purpose; Suggest adding that you use scans from one sequence to predict heating for other sequences.

The sentence now reads: “To propose a MR-thermometry method and associated data processing technique to predict the maximal RF-induced temperature increase near an implanted wire for any other MRI sequence”

R2.2: Ln 51: Is 0.5 °C maximum error, RMSE etc.? Please clarify.

Thank you for this remark. The difference between maximal predicted and measured temperature increases was around 5% for the three tested sequences.

Therefore, we corrected the sentence as follow:

“After successful fitting of temperature curves of the calibration data set, temperature rise predicted by the model was in good agreement (around 5% difference) with measured temperature by a fiber optic probe, for three other MRI sequences. “ 

R2.3: Ln 204-205: Please reformulate “The tip of the lead screwed into the myocardium was inserted…”

We reformulated the sentence as follow:

“The tip of the lead that is normally screwed into the myocardium was inserted into the gel while the other extremity was left in the air”

R2.4: Ln 225-227: What was the TR for these scans? Was it the same for all TEs (if not, compare the precision later on gets challenging)?

Repetition time was identical for each measurement and set to 1s. This is now indicated in the text.

R2.5: Ln 249-250: Does the Gaussian have the same width in all three dimensions (what was it?)? Or is it elongated like most focused ultrasound focal spots?

Yes, we assume a Gaussian source with isotropic dimensions. In our model, we do not need to explicitly provide a width for the Gaussian function, since this is included in the parameter “tau”. The idea here is that if you have a Gaussian source of heating, then you can use Eq 1 to fit the temperature curve vs time at the hottest point to derive alpha and tau (see ref 35 for details). 

In order to avoid any confusion, we removed « as for high intensity focused ultrasound » from the sentence that now reads:”

“Considering that heating induced near an implanted wire is localized around its tip, we chose to approximate this heating source by a Gaussian function with isotropic dimensions.”

R2.6: Ln 350: This reviewer can’t seem to find any blue line (showing “baseline”)? Also, consider calling it “background” rather than “baseline”.

Effectively, we forgot to include these blue lines in the figure. Instead, we added a blue arrow (as in Figure 3) in Figure 4 to show the pixel selected for background temperature evolution. We also changed “baseline” into “background” as suggested.

R2.7: Ln 408-410: It’s a bit unclear what you mean with this sentence - simply that you can use the plot in 5c to estimate the maximum temperature rise?

What is meant here is that the method relying on time-averaged power deposition are irrelevant to predict the maximal temperature rise. 

The sentence has been modified as follow to clarify this point:

“On the contrary, using the calibration curves (namely coefficient �3) from Fig 5c to estimate the maximal temperature increase from the total energy of each sequence leads to 2.78°C, 2.96°C and 2.42°C, respectively. Such an approach is thus irrelevant for predicting maximal temperature rise in the context of an implanted wire.”

R2.8: Ln 436: “(1 s refresh rate in our setting)” please add “with a 5 s temporal foot print”. This is important, as you had to shift the temporal curve to align it with the probe measurements. Hence, it is not optimally suited for real-time applications (similar to using a sliding window reconstruction for undersampled k-space data).

We added “with a 3 s temporal footprint”, not 5 s, since this is the time shift we needed to align MR-temperature curve and probe measurements. 

R2.9: Ln 448: Change “FA ≥ 20°” to “FAHM ≥ 20°”, right?

Changed

R2.10: Ln 472: This doesn’t quite tell the full story, right? This assumes a “step function” going straight to 18 °C and holding it there for 3 s. In reality you will start accumulating dose as soon as your increase is 6 °C so with the fairly slow heatings shown in Fig 4 you’ll have a substantial dose before getting to 18 °C.

These numbers were introduced in the revised manuscript to illustrate that having a 3 s latency is not considered problematic, since a substantial heating (18°C) is required to reach a lethal thermal dose within this time scale. In the discussion section, we already explained that the sequence may be used with shorter duration of the heating module, which will also reduce the risk of accumulating excessive thermal dose. Altogether, we believe that this 3 s latency is not an issue. In order to simplify the text, we propose to modify the sentence as follow:

“Despite the proposed implementation creates a latency of 3 s, the risk of reaching a lethal thermal dose within this time scale (according to the CEM43 (40) model) is unlikely to occur with optimized clinical devices.”

R2.11: Figure 8: Both the CINE and TSE predict the heating and start of cooling pretty well, but starts to deviate substantially at the end of the cooling period – can the authors speculate why this is?

This deviation can result from uncertainty in estimation of alpha and tau parameters that are used to simulate the heating. In this case, deviation may accumulate and become more visible at the end of the cooling. However, as stated in the discussion section, the relevant parameter from the IEC is the maximal temperature rise, which is predicted within 5% error with our method. Since this point is already indicated in the previous revised manuscript, we prefer to keep the text unchanged.

---

## [Editor Report · Decision Letter 2]

12 Apr 2021

A fast MR-thermometry method for quantitative assessment of temperature increase near an implanted wire

PONE-D-20-32401R2

Dear Dr. DELCEY,

We’re pleased to inform you that your manuscript has been judged scientifically suitable for publication and will be formally accepted for publication once it meets all outstanding technical requirements.

Kind regards,

Nick Todd, PhD

Academic Editor

PLOS ONE
---

## [Editor Report · Acceptance letter]

5 May 2021

PONE-D-20-32401R2 

A fast MR-thermometry method for quantitative assessment of temperature increase near an implanted wire 

Dear Dr. Delcey:

I'm pleased to inform you that your manuscript has been deemed suitable for publication in PLOS ONE. Congratulations! Your manuscript is now with our production department. 

Kind regards, 

on behalf of

Dr. Nick Todd 

Academic Editor

PLOS ONE